# Automated identification of multinucleated germ cells with U-Net

**Samuel Bell**[1,2]*, **Andras Zsom**[1], **Justin Conley**[3], **Daniel Spade**[1]

**1** Brown University, Providence, RI, United States of America, **2** Planetary Science Institute, Tucson, AZ, United States of America, **3** U.S. Environmental Protection Agency/ORD/CPHEA/PHITD/RDTB, Research Triangle Park, NC, United States of America

* sbell@psi.edu

**Data Availability Statement:** The data files can be downloaded from the Brown Digital Repository (DOI: 10.26300/rv2a-kp40). Additionally, some limited data are contained within the code repositories, which are available on GitHub at

## Abstract

Phthalic acid esters (phthalates) are male reproductive toxicants, which exert their most potent toxicity during fetal development. In the fetal rat, exposure to phthalates reduces testosterone biosynthesis, alters the development of seminiferous cords and other male reproductive tissues, and induces the formation of abnormal multinucleated germ cells (MNGs). Identification of MNGs is a time-intensive process, and it requires specialized training to identify MNGs in histological sections. As a result, MNGs are not routinely quantified in phthalate toxicity experiments. In order to speed up and standardize this process, we have developed an improved method for automated detection of MNGs. Using hand-labeled histological section images with human-identified MNGs, we trained a convolutional neural network with a U-Net architecture to identify MNGs on unlabeled images. With unseen hand-labeled images not used in model training, we assessed the performance of the model, using five different configurations of the data. On average, the model reached near human accuracy, and in the best model, it exceeded it. The use of automated image analysis will allow data on this histopathological endpoint to be more readily collected for analysis of phthalate toxicity. Our trained model application code is available for download at github.com/brown-ccv/mngcount.

## 1. Introduction

Phthalates (phthalic acid esters) are used to make a variety of industrial products and consumer goods, most notably to plasticize polyvinyl chloride for use in products such as vinyl sheeting and medical tubing [1]. Human exposure to phthalates is nearly universal, and their potential toxicity to male reproductive tract development raises concern for male reproductive health [2, 3, 4, 5, 6, recently reviewed by 7, 8]. In rat models with *in utero* exposure to certain phthalates, decreased testosterone production is observed [9, 10, 11, 12, 13, 14]. However, quantification of male reproductive toxicity is complicated by lack of concordance between this effect on steroidogenesis and other adverse testicular development outcomes, as histological effects on the testis can occur regardless of a reduction in measured testosterone [4, 3, 2, 15, 16]. One effect of phthalate exposure on fetal testis development that is consistent across

https://github.com/samwbell/train_unet_mng and
https://github.com/brown-ccv/mngcount

**Funding:** DS, R00 ES025231, National Institute of
Environmental Health Sciences, https://www.niehs.
nih.gov/ The funders had no role in study design,
data collection and analysis, decision to publish, or
preparation of the manuscript.

**Competing interests:** Justin Conley is employed by
the US Environmental Protection Agency (USEPA).
This manuscript has been internally reviewed by
the USEPA. This does not alter our adherence to
PLOS ONE policies on sharing data and materials.

species is an increase in multinucleated germ cells (MNGs), germ cells which contain two or more nuclei [9, 17]. Di-*n*-butyl phthalate and di-(2-ethylhexyl) phthalate induce MNGs in rats and mice *in utero* and in rat, mouse, and human tissue xenograft models, to levels much greater than the background rate of MNGs found in control testes [16, 2, 3, 18, 19], which has led researchers to conclude that, despite differences in the anti-androgenic response, rats are an appropriate model in which to study the effects of phthalates on germ cells [18]. The long-term impact of MNG induction on testis health is unclear. Phthalate-induced MNGs are degenerative cells that are lost through p53-dependent apoptosis in the early postnatal period [20, 21, 9, 22]. However, induction of MNGs is clearly a reproducible indicator of phthalate effect on the fetal testis. Although this is a quantitative endpoint with the potential to be used as a biomarker of seminiferous cord-mediated phthalate toxicity, thorough studies on the dose-response for induction of MNGs by most phthalates have not been conducted in any species [19].

The primary drivers of the low quantity of MNG count data are the time required for performing the count and the training needed for an expert to be able to confidently identify the MNGs [19]. Additionally, there is error inherent in recognizing an MNG in a histological section, or especially a two-dimensional image of a histological section. This is due at least in part to the high density of germ cells in seminiferous cords, the large germ cell nuclei and low ratio of cytoplasm area to nucleus area in cross-section, and the sometimes indistinct germ cell membrane. As a result of these features, different experts are able to reach different conclusions about some fraction of the cells that are identified as MNGs [19] began the process of addressing these issues through the creation of a semi-automated counting pipeline. Using hematoxylin-stained thin sections of fetal rat testes and a scripted process though the NIH ImageJ software [19], identified MNGs based on their image characteristics using computer vision principles. The primary image criteria used to identify MNGs was size. Without a formal cell segmentation routine [19], used image thresholding, down-sampling, blurring, and filling to identify connected image components. These connected components were filtered based on circularity to exclude non-cells, and then they were filtered by size to identify MNGs. While much of this pipeline was fully automated, several steps required human input, including the initial image processing and the identification of the thresholds for each image batch.

Here, we propose to improve on the [19] approach, leveraging recent innovations in convolutional neural networks to build a fully automated MNG identification code.

## 2. Methods

### 2.1 Neural network architecture

The base neural network architecture we chose was U-Net. Developed by [23], U-Net was designed for solving the cell segmentation problem. While many neural networks, when trained, take an image as an input and produce a classification probability, U-Net produces a mapped image showing the probability of a certain classification for each part of the image. For cell segmentation, U-Net produces a map of where the cell boundaries are located.

U-Net takes a 512 by 512 pixel image tile as its input. As the image tile proceeds through the U-Net, it passes through multiple convolution, max pooling, and ReLU activation function layers, resulting in an image cube with reduced x and y dimensions and enhanced z dimensions. At this layer, the deepest learning occurs. The image is then up-sampled using up-convolution, finally producing an output prediction map with values scaled from 0 to 255. The ReLU activation function, which sets all negative values to zero, helps the neural network focus in on specific areas of the image.

There are several implementations of the U-Net, and we chose a Python and Keras implementation published by GitHub user zhixuhao (https://github.com/zhixuhao/unet). While we modified the training, augmentation, prediction, and data management code extensively, we left the original network architecture unmodified.

Because it was originally designed for solving the cell segmentation problem, the original implementations of U-Net used a map of cell boundaries as the training data. The cells were outlined, not filled in. To apply the U-Net to the MNG problem, we structured the training data somewhat differently. Instead of the outline of each MNG, we used a map of the filled in area of each MNG. As a result, the trained network produced a prediction heat map with a likelihood of each pixel belonging to an MNG (Fig 1).

## 2.2 Data acquisition

Experiments involving animals were performed under a protocol approved by the Institutional Animal Care and Use Committee of the USEPA National Health and Environmental Effects Research Laboratory (Laboratory Animal Project Review #19-03-001). Animals were housed in a facility accredited by the Association for Assessment and Accreditation of Laboratory Animal Care and maintained at 20–22°C, 45–55% humidity, and a 12:12 h photoperiod (lights off at 1800 hrs). Histological section images were obtained from the samples from animals exposed to diethyl phthalate (DEP), dipentyl phthalate (DPeP), dimethyl phthalate (DMP), di-(2-ethylhexyl) phthalate (DEHP), di-n-butyl phthalate (DBP), benzyl butyl phthalate (BBP), di-(2-ethylhexyl) tetrabromophthalate (TBPH), dioctyl terephthalate (DOTP), or corn oil vehicle reported previously by [19]. An additional 15 samples were obtained from rats treated with DPeP (1, 11, 33, or 100 mg/kg/d) or corn oil vehicle as previously described. Briefly, timed pregnant Sprague Dawley rats were obtained from Charles River Laboratories (Raleigh, NC). Treatment compounds were administered by daily oral gavage in 2.5 mL/kg body weight corn oil vehicle from gestation day (GD 17–21). Dams were euthanized by decapitation approximately two hours after administration of the final dose. Testes were isolated from male fetuses and treated with a modified Davidson's fixative for 15 minutes, before being transferred to 70% ethanol.

Fixed testes were dehydrated through a series of graded ethanols, embedded in paraffin wax, and sectioned at a thickness of 5 μm. Paraffin sections were mounted on glass histological slides, deparaffinized, rehydrated, and stained with hematoxylin. Hematoxylin stained sections were scanned at 40x magnification on an ImageScope CS digital slide scanner (Leica Biosystems, Buffalo Grove, IL) and saved in the ScanScope default.svs file format.

Images were down-sampled to 1 micron per pixel and compressed using the NIH's ImageJ software, then converted to 24-bit RGB PNG files. We determined that the color ratios did not contain additional useful information beyond the grayscale values, so we converted the color images to single-channel 8-bit greyscale images. We then cropped sub-images around each individual testis slice on the slide. In most cases, each slide contained four testis slices.

## 2.3 Manual identification of MNGs

For each image, we hand-identified the centers of each MNG by using Leica ImageScope software to view digital slide images. In order to ensure consistency, hand-identification was performed by the human reference scorer, who is experienced in identification of MNGs. In order to quantify the accuracy of human identification, two additional researchers performed our own hand MNG identification on previously unseen slides. Human test scorer 1 examined 12 testes that he had not previously seen, identifying 44 MNGs. Using the reference human scorer's annotations as true MNG identifications, these 44 MNGs consisted of 10 false

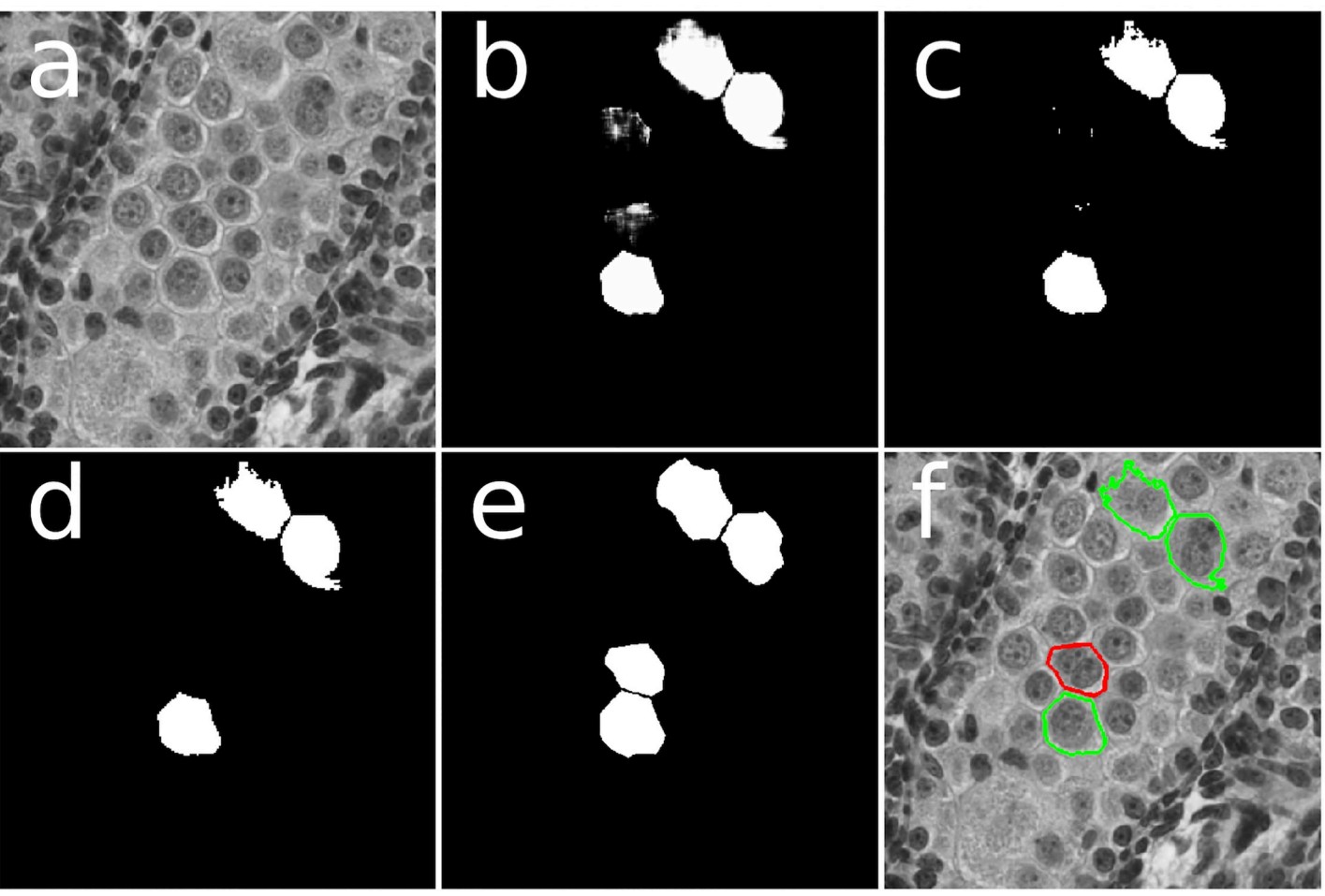

**Fig 1.** A schematic showing the different stages of the automated MNG prediction process, as performed on a single panel measuring 512x512 pixels (here 1 pixel = 1 micron): a) The original panel. b) The predictions after the trained model has been applied. (In this case, the fold configuration with Fold 0 as the holdout set is being used.) The likelihood of a pixel being represented by an MNG has been mapped to the pixel brightness values on an arbitrary 0 to 255 scale, creating an MNG location heatmap. These brightness values should be seen as relative likelihood, and the exact likelihood depends on the model. Note the clearly visible MNGs and some brighter pixels around two cells that the model sees as having a possibility of being an MNG. c) The MNG location prediction image after a brightness cutoff threshold has been applied. For this fold configuration, the brightness cutoff is 240. After applying the brightness cutoff, the two MNGs close together clearly separate, and the not clearly predicted MNGs become a few specks. d) The final prediction image after a cutoff area threshold has been applied. All shapes with an area below the cutoff, 150 pixels in this case, have been removed. (We used the Green's theorem area approximation, not literal pixel areas.) This image shows the final predictions of MNG locations. e) The "true" MNG locations as identified by the human reference scorer. f) The successful and unsuccessful predictions shown superimposed on the original image. Correctly predicted MNGs are outlined in green, and the MNG that was missed is outlined in red.

positives, and 34 true positives. There were an additional 11 false negatives. Human test scorer 2 examined fourteen images, identifying 105 MNGs, including 10 false positives, with an additional 41 false negatives in this dataset.

To quantify the match between human scorers or the model, we utilized the F1 score metric, which is the harmonic mean of the precision and the recall:

$$precision = \frac{true\ positives}{true\ positives + false\ positives}$$

$$recall = \frac{true\ positives}{true\ positives + false\ negatives}$$

$$F1 = \left( \frac{precision^{-1} + recall^{-1}}{2} \right)^{-1},$$

where the true positives are MNGs that were successfully identified; the false negatives are MNGs that were identified by the reference human scorer, but not by the human test scorers or the model; and the false positives are MNGs identified by the human test scorers or the model but not the reference human scorer. For the purposes of these statistics, we treated the reference human scorer's identifications as the true identifications, but we understand that likely they are not perfectly accurate. The F1 scores for human test scorers one and two were 0.764 and 0.788 respectively.

To construct the true data maps, we used the reference human scorer's identifications. After the identifications had been made, we manually identified the outlines of the MNG cells, drawing filled polygons.

## 2.4 Augmentation and tile selection procedure

In order to increase the number of images on which the network is trained, the training set for a neural network is typically augmented by passing the true maps and raw images through a series of transformations. For each MNG, we selected 500 sample panels with the MNG center located randomly within the panel. As a result, we were able to sample the MNG with a large number of randomized horizontal and vertical displacements. For each sample panel, we iterated through all eight unique combinations of vertical and horizontal flips and 90˚C and 180˚C rotations. By randomizing the orientation and location of the MNGs, we ensured that the network would not falsely train on features related to the orientation and location of the MNG. Randomizing the location also increased the area of negatives to train the model on.

## 2.5 Training process

We separated the data into five separate "folds" or batches of images. For the training set, we used three of the folds, with one fold each for the test and holdout sets. The training set was used for training the model, the test set was used for optimizing parameters, and the holdout set was used for assessing the accuracy of the model. We used five different combinations of these folds, with each fold being used once as the holdout set. In order to ensure that the folds were both representative and fully independent from each other, we used batch stratification. There were 28 slides, with four testis slices on most slides. To ensure representative batches, we sorted the slides by the number of MNGs on each slide and separated them into five groups of five and one group of three, ordered by number of MNGs. We assigned each slide from every group to a fold, making sure than no fold received more than one slide from each group and selecting the configuration that minimized the variation in the total number of MNGs in each fold. Unfortunately, we could not fully equalize the number of MNGs in each fold. One slide in Fold 0 had 152 MNGs, more than the average number of MNGs per fold, 119.2. In order to accomodate this outlier slide, Fold 0 had to have 175 MNGs, substantially more than the other folds, which ranged from 103 to 108 MNGs.

Numbering the five folds 0–4, we ran the model on five configurations. In each case, we selected the test set to be the fold numerically after the holdout set fold number (looping back to fold 0 for the test set when the holdout set was fold 4), with the other three folds going into the training set. We used the data from the training set to train the model, running it for fifteen epochs and saving the results from each epoch. For each epoch, we pass the model through the

full set of augmented training data, taking the output of the previous epoch as the starting model for the next one.

The result of the trained neural network is a map of MNG probability for each pixel scaled as a brightness with integer values ranging from 0 to 255. To convert it into a binary map of MNG locations, we used brightness and area cutoffs (Fig 1). All pixels below the brightness cutoff were removed, and only remaining connected regions of pixels with areas of above the area cutoff were retained. For reasons of computational efficiency, the areas we used here were Green's theorem areas, which are typically slightly lower than literal pixel areas. To calculate the optimal brightness and area cutoffs, as well as the optimal epoch, we used a grid search, selecting the combination of values these three parameters that maximized the F1 score (Figs 2 and 3). With the epoch, brightness cutoff, and area cutoff optimized and selected, we arrived at a final model that could take an image and predict the locations of the MNGs. To measure the final accuracy, we tested this model on the holdout set and measured the F1 score. By using these separate training, test, and holdout sets, we avoided overfitting, which would distort our accuracy scores.

## 3. Results

We successfully ran our model on all five of the fold configurations. There was little consistency in the optimal parameter values among the different fold configurations. There was especially great variety in the optimal epoch, with values ranging from 1 to 13 (Table 1). The fold configuration with Fold 4 as the holdout set led to the lowest holdout set F1 score, 0.622, and its optimal cutoff area was a rather extreme outlier of 20 pixels. This fold combination used Fold 0, which had a much larger number of MNGs than the other folds, as its test set. The fold combination with the second lowest holdout set F1 score was when Fold 0 was the holdout set, and in that case the optimal cutoff area was 150 pixels, above the outlier of 20 pixels but still clearly lower than for the other folds. When Fold 0 was not part of the training set, the model had fewer MNGs to train on, so it makes sense that the performance would be worse.

Our best holdout set F1 score was 0.805, coming in the fold configuration with Fold 3 as the holdout set (Table 2). This exceeded the accuracy of either human scorer, whose F1 scores were 0.764 and 0.788. However, the mean holdout set F1 score across all five fold configurations was 0.724, with a standard deviation of 0.07, falling slightly below the scores for the human scorers. On average, the model reaches near human accuracy, and in the best model, it exceeds it.

In all except the best performing fold configuration, false negatives exceeded false positives, with an average false positive to false negative ratio of 0.62. However, in the best performing model false positives exceeded false negatives 27 to 17, which corresponds to a false positive to false negative ratio of 1.59. For all the folds, the average false positive to false negative ratio was 0.81.

## 4. Discussion

Ultimately, the strongest underlying limitation we faced was the training data. The accuracy of a neural network is only as good as the training data, and it can always be improved with more training data. Our model is no exception. We can see this clearly in our data because the fold configurations without Fold 0 and its especially high MNG count in the training set underperformed compared to the three fold configurations that had Fold 0 in the training set. More training data would certainly improve the model.

However, more data may not be enough to reach perfect accuracy. In addition to the limited quantity of training data, we were also limited by error inherent in the scoring procedure

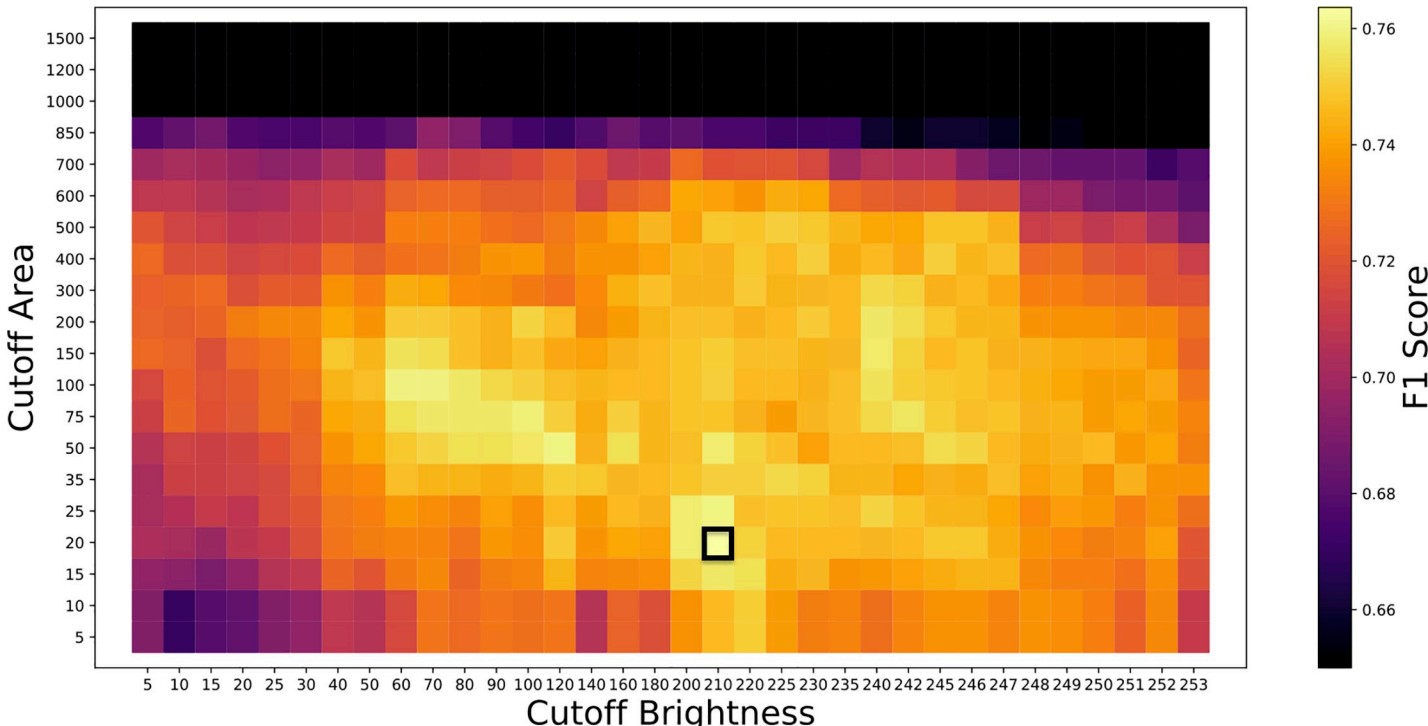

**Fig 2. An example of the grid search over the test set to find the optimal cutoff area, cutoff brightness, and F1 score.** This example shows the fold configuration with Fold 4 as the holdout set and Fold 0 as the test set. Here, F1 score is plotted for every cutoff brightness and cutoff area we simulated. We selected the parameter combination that maximized the F1 score, a cutoff area of 20 pixels and a cutoff brightness of 210, and we have indicated this particular grid point on the figure it with a black box around it. The f1 score color bar was scaled between a minimum of 0.65 and the maximum brightness, with f1 scores below 0.65 set to black. The cutoff area and cutoff brightness parameters convert the output MNG location probability map, scaled as a brightness from 0 to 255, to specific MNG location maps. First the brightness cutoff is applied, and then any contiguous regions of pixels above the cutoff smaller than the cutoff area are removed.

used to obtain training data. There are several issues that make manual MNG identification from a single slide difficult. The thin sections on each slide merely contain 2-D cross-sections of 3-D cells. While some MNGs can be clearly identified on a slide because multiple nuclei are clearly visible, in many cases the cross section does not slice through all the nuclei. In some cases, the second nucleus is only faintly visible, and in others it may not be visible at all, even though part of the cell is visible on that cross-section. In other cases, two nuclei may be very close together, and it may not be clear whether they are separated by cell membranes or not. Many cells contain localized hematoxylin-stained cytoplasmic regions that cannot be easily distinguished from the edge of a nucleus, even by a highly trained eye.

To develop the most robust possible set of training data would require serial sectioning, a pseudo-3D approach. Under this approach, three serial thin sections would be taken from each sample, with the top and bottom layers informing the identification of MNGs on the middle layer, as in [4]. In this approach, scoring is performed in the central section. Ambiguous MNG identifications in the central section can often be confirmed on an adjacent section, because MNGs are significantly larger than the 5 μm diameter of the sections. When an MNG can be clearly identified in the top or the bottom layer but is unclear in the middle layer, then we should be able to more confidently identify it in the middle layer. This way, we should be able to build up a much more accurate set of training data. With those more accurate training data, we should be able to train a model that may well exceed human accuracy.

The error inherent in human modeling is such that our "true" labels produced by the human reference scorer are almost certainly at least partially inaccurate. When the model

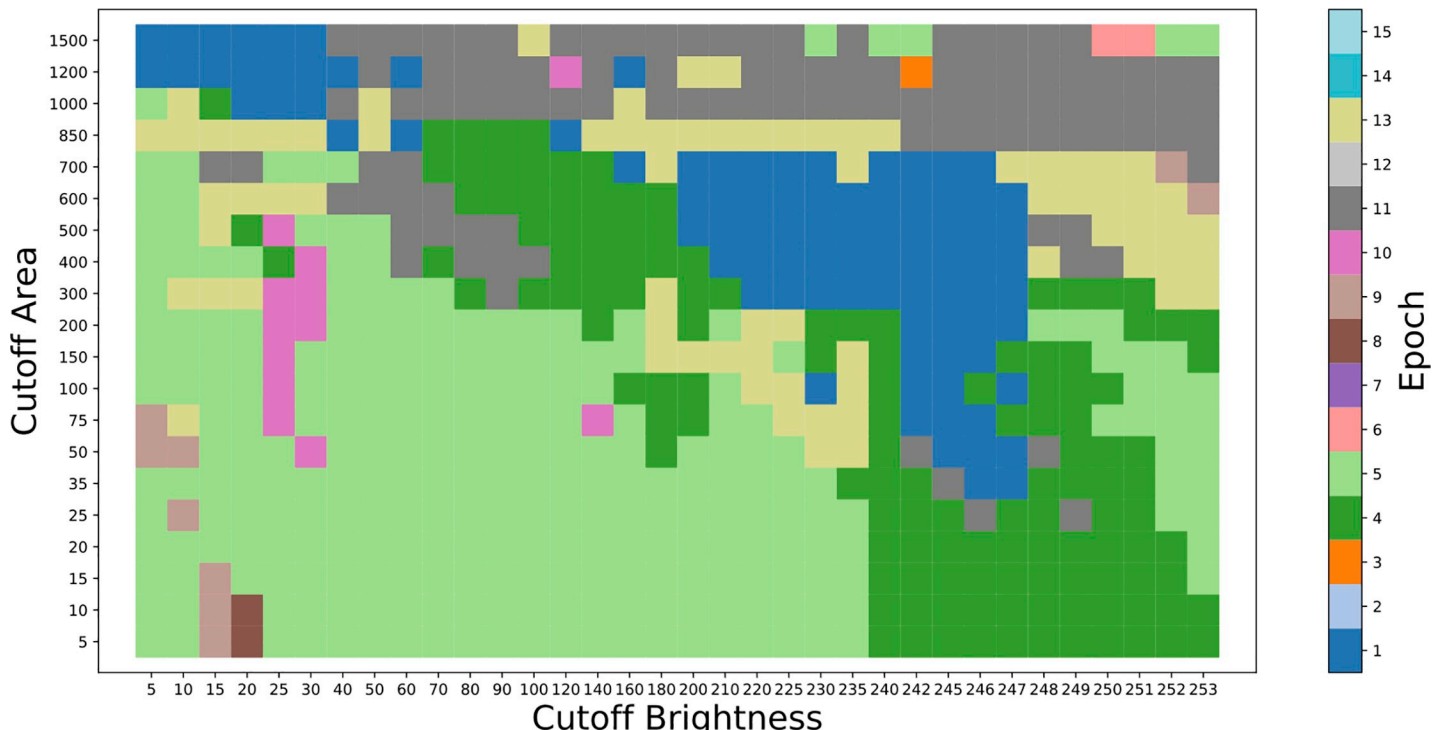

**Fig 3. The grid search shown in Fig 2, but with epoch instead of F1 score plotted.** The cutoff area and cutoff brightness parameters convert the output MNG location probability map, scaled as a brightness from 0 to 255, to specific MNG location maps. First the brightness cutoff is applied, and then any contiguous regions of pixels above the cutoff smaller than the cutoff area are removed.

differs from the "true" labels, it is always possible that the model is correct, and the "true" labels are not. Additionally, human scorers may be more likely to err in similar ways, and the F1 scores between the scores of human scorers may be somewhat elevated for this reason. For instance, human scorers may be unusually prone to miss an MNG, compared to a trained model that will always examine all of the image.

We also note that our model used the same laboratory to produce and process all the slides. A different group reproducing these results might well wind up with images with slightly different treatment, lighting, and end image conditions. We cannot ensure that our model will achieve the same level of accuracy on images produced by other groups in other laboratories. We have, in a sense, overfit the model to the particular researchers and set-up used to produce the images we have analyzed. By training the model with a much wider array of researchers and laboratory set-ups, this issue could be ameliorated through future work.

**Table 1. Results of the grid search to optimize the epoch, cutoff brightness, and cutoff area.**

| Holdout set fold | Test set fold | Optimal Epoch | Optimal Cutoff Brightness | Optimal Cutoff Area | Test Set Matches | Test Set False Positives | Test Set False Negatives | Test Set F1 Score | Holdout Set F1 Score |
|---|---|---|---|---|---|---|---|---|---|
| 0 | 1 | 13 | 240 | 150 | 78 | 22 | 29 | 0.754 | .710 |
| 1 | 2 | 2 | 120 | 700 | 78 | 4 | 25 | 0.843 | .714 |
| 2 | 3 | 1 | 246 | 400 | 94 | 17 | 14 | 0.858 | .770 |
| 3 | 4 | 5 | 220 | 300 | 73 | 17 | 30 | 0.756 | .805 |
| 4 | 0 | 5 | 210 | 20 | 126 | 29 | 49 | 0.764 | .622 |

**Table 2. Holdout set results for the five folds as well as the two human scorers.**

|  | Matches | False Positives | False Negatives | Holdout Set F1 |
|---|---|---|---|---|
| Best (Fold 3) | 91 | 27 | 17 | .805 |
| Fold 0 | 119 | 41 | 56 | .710 |
| Fold 1 | 71 | 21 | 36 | .714 |
| Fold 2 | 72 | 12 | 31 | .770 |
| Fold 4 | 61 | 32 | 42 | .622 |
| Mean |  |  |  | 0.724 |
| Std. Dev. |  |  |  | 0.070 |
| Human 1 | 34 | 10 | 11 | 0.764 |
| Human 2 | 95 | 10 | 41 | 0.788 |

We were careful to do everything we could to minimize this sort of overfitting. One crucial step we took was to ensure that each slide would fall in its own fold. We did not split up multiple testes on a single slide and place them in different folds. As a result, we controlled against overfitting to the particular conditions of an individual slide, such as lighting. Had we split up the slides, we could have achieved much more equal folds, without large variability in the total number of MNGs and the types of MNGs in each fold.

The resulting variability among the folds is a major driver of error in accurately assessing the F1 scores. Even among folds with similar MNG counts, some folds will be easier for the model to count than others. For instance, in the model with the best holdout set score, where Fold 3 is the holdout set, the holdout set score, 0.805, is higher than the test set score, 0.756. In all the other cases, the holdout set score is lower than the test set score, which makes sense because the model in the test set has been optimized to produce the best possible F1 score for the test set. So the high F1 score when Fold 3 is the holdout set may be partially driven by Fold 3 being easier to count. When Fold 3 is used as the test set, it produces a slightly higher test set F1 score than any other fold configuration, which is additional evidence that Fold 3 may simply be easier for the model to count.

## 5. Conclusions

The MNG identification problem is a difficult task for humans to perform consistently. We have shown that a convolutional neural network using the U-Net architecture can approach near human accuracy and, in the case of the best model, exceed it. This new automated approach is significantly faster and involves much less human input, which will facilitate the generation of dose-response data for induction of MNGs by phthalates. The code for applying the trained model can be downloaded from github.com/brown-ccv/mngcount, and the code for training the model can be found at github.com/samwbell/train_unet_mng.

## Acknowledgments

We would like to thank Christy Lambright and Earl Gray from the Environmental Protection Agency for conducting the rat phthalate exposures and providing the tissue samples used in this study. We would like to thank Sklyar Loeb and Rebka Ephrem for their assistance with cell boundary identification during the creation of the training data. The manuscript has been subjected to review by the U.S. EPA Center for Public Health and Environmental Assessment and approved for publication, but the views expressed do not necessarily reflect the views or the policy of the U.S. EPA. Part of this research was conducted using computational resources and services at the Center for Computation and Visualization, Brown University.

## Author Contributions

**Conceptualization:** Samuel Bell, Andras Zsom.

**Data curation:** Samuel Bell, Daniel Spade.

**Formal analysis:** Samuel Bell, Andras Zsom.

**Funding acquisition:** Daniel Spade.

**Investigation:** Samuel Bell, Justin Conley, Daniel Spade.

**Methodology:** Samuel Bell, Andras Zsom, Justin Conley.

**Project administration:** Samuel Bell, Andras Zsom, Daniel Spade.

**Software:** Samuel Bell.

**Supervision:** Samuel Bell, Daniel Spade.

**Validation:** Samuel Bell, Justin Conley.

**Visualization:** Samuel Bell.

**Writing – original draft:** Samuel Bell.

**Writing – review & editing:** Samuel Bell, Andras Zsom, Justin Conley, Daniel Spade.

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
