## [Decision Letter · Decision Letter 0]

11 Mar 2020

PONE-D-20-04638

Automated identification of multinucleated germ cells with U-Net

PLOS ONE

Dear Dr. Bell,

Thank you for submitting your manuscript to PLOS ONE. After careful consideration, we feel that it has merit but does not fully meet PLOS ONE’s publication criteria as it currently stands. Therefore, we invite you to submit a revised version of the manuscript that addresses the points raised during the review process.

This paper describes an automated process to determine the number of multinucleated cells in testicular tissue sections. I agree with the reviewers that the study is of importance but also support their critical views on some details. The authors need to deal with the very thoughtful comments of the reviewers, provide adequate responses and revise the paper accordingly.

We would appreciate receiving your revised manuscript by Apr 25 2020 11:59PM. To enhance the reproducibility of your results, we recommend that if applicable you deposit your laboratory protocols in protocols.io, where a protocol can be assigned its own identifier (DOI) such that it can be cited independently in the future. For instructions see: http://journals.plos.org/plosone/s/submission-guidelines#loc-laboratory-protocols

We look forward to receiving your revised manuscript.

Kind regards,

Stefan Schlatt

Academic Editor

PLOS ONE

Journal Requirements:

"Justin Conley is employed by the US Environmental Protection Agency (USEPA).  This manuscript has been internally reviewed by the USEPA.".

Reviewers' comments:

Reviewer's Responses to Questions

**Comments to the Author**

1. Is the manuscript technically sound, and do the data support the conclusions?

Reviewer #1: Yes

Reviewer #2: Yes

2. Has the statistical analysis been performed appropriately and rigorously? 

Reviewer #1: N/A

Reviewer #2: I Don't Know

3. Have the authors made all data underlying the findings in their manuscript fully available?

Reviewer #1: Yes

Reviewer #2: Yes

4. Is the manuscript presented in an intelligible fashion and written in standard English?

Reviewer #1: Yes

Reviewer #2: Yes

5. Review Comments to the Author

Reviewer #1: The manuscript "Automated identification of multinucleated germ cells with U-Net" describes the establishment and validation of a neural net with a U-Net architecture to identify multinucleated germ cells (MGN). MGN are formed due to exposure of phthalates. Phthalates are reproductive toxicants that derive from plastic softeners and are most likely responsible for a diminished fertility, also given from mother to fetus during pregnancy. As MGN are not easy to identify, an automated system is needed. The authors describe in a detailed way the establishement, training and validation of the U-Net neural net. There, they also describe the limitations of the system and how to overcome these. As a result, the U-Net is found to be as good as the human reviewer in most cases.

Minor points of criticism:

Two references (e.g. Kavlock et al. 2006 and Ronneberger et al. 2015) are not cited in the reference list.

Isn't citing in Plos ONE performed using numbers in the text?

Reviewer #2: The manuscript aims to design and test an automated system to quantify multinucleated gonocytes (MNG) in testicular tissue sections. The study concludes that the automated program performed close to human accuracy.

General Comments

Overall the study appears technically sound. Methods to improve accuracy and efficiency of quantification of cell types are important to establish. The main areas for improvement relate to the need for a clear definition of what constitutes an MNG and their biological importance. In addition, the interpretation of the results and the utility of the system for obtaining accurate numbers of MNGs in practice should be developed. The ‘gold standard’ used to determine the accuracy of human and automated counting is the opinion of one experienced observer rather than an objective measure. Whilst the authors report that the models perform close to human accuracy, the human quantification performed relatively poorly compared to the ‘gold standard’ in terms of a high number of false negatives and false positives.

Specific Points

Page 9 Line 40 – ‘potential toxicity’ or cite direct evidence for effects of phthalates on human reproductive tract

Page 9 Line 42 – mentioned critical window in abstract. No mention of the MPW here

Page 9 Line 49 – there is no definition of an MNG. How many nuclei? How are they distinguished from a mitotic cell

Page 9 Line 50 – in which species?

Page 9 Line 52 – There is no mention of the biological importance of MNG. Given that MNGs are also identified in normal testis tissue the relevance of these should be described. Worth commenting on differences in MNG/GC aggregation between rodent and human (e.g. van den Driesche, Environmental Health Perspectives, 2015)

Page 11 Line 98 – provide a link or citation to this implementation

Page 12 Line 116 – explain how the pixel brightness indicates an MNG

Page 13 Line 134 – provide ethics reference number for the animal work

Page 16 Line 214 – how were the 3 folds for the training set chosen

Page 18 Line 258 – it is not clear how this relates to the values on the figures?

Page 19 Line 280 – what are the units of measurement for the ‘20’

Page 19 Line 282 – explain the significance of this finding

Page 20 Table 1 – final column ‘0.710’ etc

Page 20 Line 294 – how much more accurate is the model for this holdout set?

Page 21 Line 297 – the wording here is confusing and difficult to interpret

Page 21 Line 299 – why is this important? Why is it better to have more false negatives than positives? Is it not more relevant to say how many incorrect classifications there were?

Page 21 Table 2 – the number of incorrect classifications is still very high. There are 32-46% incorrect classifications in the five folds compared with ~35% incorrect classifications in the human data. For discussion on whether any of the methods are accurate enough

Page 22 Line 309 – how are the authors planning to improve the training data?

Page 23 Line 333 – why has the additional training data not been performed?

Page 23 Line 336 – is there a possibility of having more than one ‘expert’ to determine consistency and agreement for ‘gold standard’

Page 24 Line 351 – discuss whether there are other ways of identifying MNGs e.g. other techniques or cellular products etc

Page 25 Line 378 – there is no definitive conclusion on whether human assessment or the new system are accurate methods for quantification of MNGs. This makes it difficult for the reader to know how to apply the findings to their own research

6. PLOS authors have the option to publish the peer review history of their article (what does this mean?). If published, this will include your full peer review and any attached files.

Reviewer #1: No

Reviewer #2: No

---

## [Author Response · Author response to Decision Letter 0]

18 May 2020

We have uploaded a specific response to reviewers document, where our responses are color-coded to differentiate them from the reviewers' comments. We believe that document is more readable.

However, we are pasting its text here, with our responses indicated by ">>>" instead of color-coding:

Reviewer #1: The manuscript "Automated identification of multinucleated germ cells with U-Net" describes the establishment and validation of a neural net with a U-Net architecture to identify multinucleated germ cells (MGN). MGN are formed due to exposure of phthalates. Phthalates are reproductive toxicants that derive from plastic softeners and are most likely responsible for a diminished fertility, also given from mother to fetus during pregnancy. As MGN are not easy to identify, an automated system is needed. The authors describe in a detailed way the establishement, training and validation of the U-Net neural net. There, they also describe the limitations of the system and how to overcome these. As a result, the U-Net is found to be as good as the human reviewer in most cases.

Minor points of criticism:

Two references (e.g. Kavlock et al. 2006 and Ronneberger et al. 2015) are not cited in the reference list.

>>>Added them. We do apologize for this omission.

Isn't citing in Plos ONE performed using numbers in the text?

>>>We do apologize for not using numbers in the text, and we have gone back and added them in.

Reviewer #2: The manuscript aims to design and test an automated system to quantify multinucleated gonocytes (MNG) in testicular tissue sections. The study concludes that the automated program performed close to human accuracy.

General Comments

Overall the study appears technically sound. Methods to improve accuracy and efficiency of quantification of cell types are important to establish. The main areas for improvement relate to the need for a clear definition of what constitutes an MNG and their biological importance. In addition, the interpretation of the results and the utility of the system for obtaining accurate numbers of MNGs in practice should be developed. The ‘gold standard’ used to determine the accuracy of human and automated counting is the opinion of one experienced observer rather than an objective measure. Whilst the authors report that the models perform close to human accuracy, the human quantification performed relatively poorly compared to the ‘gold standard’ in terms of a high number of false negatives and false positives.

>>>Both in this paper and in general, an MNG is defined as any germ cell with more than one nucleus in shared cytoplasmic space. We assume that the reviewer means that the performance of non-expert scorers against the expert scorer (“gold standard”) was relatively poor. We agree and would argue that the poor performance of humans on this task is part of the motivation for this research. Because MNG identification is a problem that humans struggle at, automation is both more difficult and more necessary. Further, as with any assessment of pathology, there will be some error inherent in identification of histological features, even when performed by an expert.

Specific Points

Page 9 Line 40 – ‘potential toxicity’ or cite direct evidence for effects of phthalates on human reproductive tract

>>>We have added the word “potential” to reflect the uncertainty about effect levels in humans. However, given that there is considerable evidence of toxicity in experimental models using human tissue and in the epidemiological literature, we have added additional citations to support this claim.

Page 9 Line 42 – mentioned critical window in abstract. No mention of the MPW here

>>>The critical window is not really essential to this study, so we have removed the mention of it in the abstract.

Page 9 Line 49 – there is no definition of an MNG. How many nuclei? How are they distinguished from a mitotic cell

>>>Added a clarification that MNGs are defined as containing two or more nuclei. MNGs do not display the chromatin condensation that is observed during mitosis. In this study, samples were obtained on gestation day 21, which falls within the quiescent period when germ cells are mitotically inactive (described in Culty et al. 2013. Biol Reprod 89:46). We have also previously shown that the sensitive window for MNG induction begins on GD 18, coincident with the initiation of the quiescent period, and that MNGs form through a non-proliferative mechanism, presumably the collapse of intercellular bridges (Spade et al. 2015. Biol Reprod 93:110).

Page 9 Line 50 – in which species?

>>>This was altered to clarify that few studies have quantified the dose-response for MNG induction by most phthalates in any species. The next paragraph expands on the reasons for this, which include the technical difficulty and time required to count MNGs. 

Page 9 Line 52 – There is no mention of the biological importance of MNG. Given that MNGs are also identified in normal testis tissue the relevance of these should be described. Worth commenting on differences in MNG/GC aggregation between rodent and human (e.g. van den Driesche, Environmental Health Perspectives, 2015)

>>>We have added a discussion on the biological significance of MNGs and their induction by phthalates in rats, mice and humans. Thank you for the suggested reference.

Page 11 Line 98 – provide a link or citation to this implementation

>>>Link added.

Page 12 Line 116 – explain how the pixel brightness indicates an MNG

>>>Added some clarifications. This is a bit difficult to explain because there is not explicit formula. The model is set up to try to reproduce the binary training images, where the pixels covered by MNGs have a pixel value of 255, and the pixels not covered by MNGs have a pixel value of 0. In the output map after an image has been run through the trained model, the closer the pixel value is to 255 the likelier it is that it is covered by an MNG, but there is no exact formula. It is specific to the model, and because neural networks are black boxes, we don’t know the exact way the model is producing it or what exactly the model means by pixel brightness.

Page 13 Line 134 – provide ethics reference number for the animal work

>>>This has been updated to indicate that the animal work was performed under Laboratory Animal Project Review #19-03-001 at the USEPA National Health and Environmental Effects Research Laboratory.

Page 16 Line 214 – how were the 3 folds for the training set chosen

>>>This is explained below in some detail. Essentially, because there are five folds, the selection of a holdout and test set leaves only three folds left over for the training set. We chose to identify the fold configurations by which fold was used for the holdout set.

Page 18 Line 258 – it is not clear how this relates to the values on the figures?

>>>We have indicated these values on the figure with a black box.

Page 19 Line 280 – what are the units of measurement for the ‘20’

>>>The unit of measurement is pixels. We have added a clarification.

Page 19 Line 282 – explain the significance of this finding

>>>Added a brief sentence explaining this.

Page 20 Table 1 – final column ‘0.710’ etc

>>>We’re not quite sure how to interpret this comment. 

Page 20 Line 294 – how much more accurate is the model for this holdout set?

>>>If this is asking how much more accurate the best model is than the human scorers, it had a holdout score of 0.805 compared to human scorer values of 0.764 and 0.788. We may not fully understand the question being asked here. We do apologize.

Page 21 Line 297 – the wording here is confusing and difficult to interpret

>>>We have restructured this as false positive to false negative ratios instead of percentages. We hope this will be less confusing and easier to interpret.

Page 21 Line 299 – why is this important? Why is it better to have more false negatives than positives? Is it not more relevant to say how many incorrect classifications there were?

>>>It’s not inherently better to have more false negatives than false positives, but it is important because it describes how the model fails. For instance, if you have an application where you are more worried about false negatives, then you probably want a model where false negatives are less common than false positives. Maybe the goal is to make sure you don’t miss anything. Plenty of machine learning algorithms are tuned in order to make many false positives but minimize false negatives because the goal is as a screening before a final pass human check. Alternatively, there are models where the goal is to make sure you don’t make a false identification. Maybe you want to make sure that you don’t falsely classify a testis as having an MNG when it doesn’t. The F1 score incorporates both into one statistic.

Page 21 Table 2 – the number of incorrect classifications is still very high. There are 32-46% incorrect classifications in the five folds compared with ~35% incorrect classifications in the human data. 

>>>This is true. The mean model performs slightly worse than the average human scorer, but the best model outperforms the average human scorer. However, to even approach human accuracy we consider a good result. The reason why there is a high rate of incorrect classifications is that this is a difficult task even for humans.

For discussion on whether any of the methods are accurate enough

Page 22 Line 309 – how are the authors planning to improve the training data?

>>>We discuss this in the next two paragraphs. We believe that more accurate training data could be obtained by using three serial thin sections. By using the central section for scoring and the sections above and below the central section to confirm MNG identifications, we believe we could have much more accurate human identification of MNGs. This approach is described in the methods of Spade et al. (2014. Toxicol Sci 138:148). It would be time consuming and laborious to use for the quantity of samples included in the current paper. However, it would provide more accurate training data, which we believe we could build a much better model. In this case, the main limitation on the training data is not so much the quantity as the accuracy.

Page 23 Line 333 – why has the additional training data not been performed?

>>>This work would require additional funding. We hope to perform this in the future.

Page 23 Line 336 – is there a possibility of having more than one ‘expert’ to determine consistency and agreement for ‘gold standard’

>>>We did not have enough collaborators on this specific project to do this, but it should be possible in theory through future work. There is a limited community of experts, but perhaps one approach could be to have a broad panel of experts reach consensus on the identifications through discussion and a voting procedure. Such an approach would require extensive collaboration and logistics, but it would be a sensible direction for future research.

Page 24 Line 351 – discuss whether there are other ways of identifying MNGs e.g. other techniques or cellular products etc

>>>There is no published method of identifying MNGs aside from identification in histological sections. The only technique we are aware of for increasing the accuracy of the count is incorporating serial thin sections in the scoring procedure.

Page 25 Line 378 – there is no definitive conclusion on whether human assessment or the new system are accurate methods for quantification of MNGs. This makes it difficult for the reader to know how to apply the findings to their own research

>>>We do not attempt to downplay the accuracy issues with human quantification of MNGs. Indeed, they are a primary motivating factor of this research. The ultimate goal is to develop a methodology that can outperform humans, and we believe we have made major progress in that direction. Despite the issues with accurately quantifying the MNG count, MNG counts are still valuable tools in assessing phthalate toxicity. We also believe it is likely that the current model could produce data of sufficient quality for dose-response assessment.

---

## [Decision Letter · Decision Letter 1]

1 Jun 2020

Automated identification of multinucleated germ cells with U-Net

PONE-D-20-04638R1

Dear Dr. Bell,

We are pleased to inform you that your manuscript has been judged scientifically suitable for publication and will be formally accepted for publication once it complies with all outstanding technical requirements.

With kind regards,

Stefan Schlatt

Academic Editor

PLOS ONE

Additional Editor Comments (optional):

Reviewers' comments:

Reviewer's Responses to Questions

**Comments to the Author**

1. If the authors have adequately addressed your comments raised in a previous round of review and you feel that this manuscript is now acceptable for publication, you may indicate that here to bypass the “Comments to the Author” section, enter your conflict of interest statement in the “Confidential to Editor” section, and submit your "Accept" recommendation.

Reviewer #1: All comments have been addressed

Reviewer #2: (No Response)

2. Is the manuscript technically sound, and do the data support the conclusions?

Reviewer #1: (No Response)

Reviewer #2: Yes

3. Has the statistical analysis been performed appropriately and rigorously? 

Reviewer #1: (No Response)

Reviewer #2: I Don't Know

4. Have the authors made all data underlying the findings in their manuscript fully available?

Reviewer #1: (No Response)

Reviewer #2: Yes

5. Is the manuscript presented in an intelligible fashion and written in standard English?

Reviewer #1: (No Response)

Reviewer #2: Yes

6. Review Comments to the Author

Reviewer #1: (No Response)

Reviewer #2: The authors have addressed the reviewer comments. There are a few instances of spelling and grammar to address and the numbers in the final column of Table 2 should be corrected e.g. '0.710' instead of '.710'

7. PLOS authors have the option to publish the peer review history of their article (what does this mean?). If published, this will include your full peer review and any attached files.

Reviewer #1: No

Reviewer #2: No

---

## [Editor Report · Acceptance letter]

26 Jun 2020

PONE-D-20-04638R1 

Automated identification of multinucleated germ cells with U-Net 

Dear Dr. Bell:

I'm pleased to inform you that your manuscript has been deemed suitable for publication in PLOS ONE. Congratulations! Your manuscript is now with our production department. 

Kind regards, 

on behalf of

Dr. Stefan Schlatt 

Academic Editor

PLOS ONE